# Search for $^{22}$Na in novae supported by a novel method for measuring femtosecond nuclear lifetimes

Classical novae are thermonuclear explosions in stellar binary systems, and important sources of $^{26}$Al and $^{22}$Na. While γ rays from the decay of the former radioisotope have been observed throughout the Galaxy, $^{22}$Na remains untraceable. Its half-life (2.6 yr) would allow the observation of its 1.275 MeV γ-ray line from a cosmic source. However, the prediction of such an observation requires good knowledge of its nucleosynthesis. The $^{22}$Na$(p, γ)^{23}$Mg reaction remains the only source of large uncertainty about the amount of $^{22}$Na ejected. Its rate is dominated by a single resonance on the short-lived state at 7785.0(7) keV in $^{23}$Mg. Here, we propose a combined analysis of particle-particle correlations and velocity-difference profiles to measure femtosecond nuclear lifetimes. The application of this method to the study of the $^{23}$Mg states, places strong limits on the amount of $^{22}$Na produced in novae and constrains its detectability with future space-borne observatories.

Nuclear reactions between charged particles in astrophysical environments proceed by quantum tunneling. The measurement of these reactions in the laboratory is very difficult because of the very small cross sections ($σ \lesssim 1$ nb). To date, about 10 reactions involving radioactive nuclei and charged particles have been directly measured at low energies. Among them, the $^{22}$Na$(p, γ)^{23}$Mg reaction has a direct impact on the amount of radioactive $^{22}$Na produced in novae[1–5]. The astrophysical relevance of $^{22}$Na for the diagnosis of nova outbursts was first mentioned by Clayton and Hoyle[6], in the context of its decay into a short-lived excited state of $^{22}$Ne, and the subsequent emission of a γ-ray photon of 1.275 MeV released during its de-excitation. With the half-life of $^{22}$Na being much longer than the transition time from an optically thick to an optically thin ejecta of a classical nova (about one week), the flux of the 1.275 MeV γ-ray line will remain close to its maximum value for months after the ejecta has become transparent to γ rays[7]. A precise determination of the $^{22}$Na$(p, γ)^{23}$Mg thermonuclear rate is necessary to improve the predicted abundances of nuclei in the mass region $A \geq 20$ during nova outbursts, including the $^{22}$Na abundance in the ejecta that impacts in turn the predicted $^{20}$Ne/$^{22}$Ne ratios in presolar grains[8] of a putative nova origin, as well as, the corresponding γ-ray emission flux. The rate of this reaction is mainly dictated by a single resonance in $^{23}$Mg at 7785.0(7) keV[5]. A direct

measurement of its strength ($ωγ$) has been performed in three different studies with conflicting results: $ωγ = 5.7^{+1.6}_{-0.9}$[5], $1.8(7)$[9] and $<0.36$ meV[10]. Indirect experimental methods, such as lifetime measurements, have also been employed to determine the strength of this resonance. Our simulations[11] show that, in the range of the debated values of the resonance strength, the mass of $^{22}$Na ejected in novae depends on the lifetime ($τ$) of this key state approximately as $M_{\text{ejec}} \propto τ^{0.7}$. The lifetime of this state was previously measured to be $τ = 10(3)$ fs[4], consistent with the recently determined upper limit of 12 fs[12]. Yet, this is at odds with the predictions of the nuclear Shell Model (SM), $τ_{\text{SM}} \approx 1$ fs.

Here we propose an experimental method for measuring short ≈fs lifetimes. This method has been applied to the key state in $^{23}$Mg in order to obtain an independent measurement and, so, to reduce systematic uncertainties. As a result, the rate of the $^{22}$Na$(p, γ)^{23}$Mg reaction is now well known, and realistic estimates of the maximum detectability distance of novae in 1.275 MeV γ rays are derived.

## Results

The experiment was performed at GANIL, France. A $^{24}$Mg beam was accelerated to 110.8(4) MeV and impinged on a $^{3}$He target of $≈2 × 10^{17}$ atoms cm$^{-2}$ uniformly implanted up to a depth of 0.1 μm below the

✉ e-mail: chloe.fougeres@gmail.com; francois.oliveira@ganil.fr

surface of a 5.0(5) μm gold foil, producing the $^{23}$Mg nucleus by the $^{3}$He($^{24}$Mg,$^{4}$He)$^{23}$Mg reaction. Both $^{24}$Mg and $^{23}$Mg nuclei were then stopped in a 20.0(5) μm gold foil. The two gold foils were mounted back-to-back. About 20 states were populated in $^{23}$Mg with excitation energies between 0 and 8 MeV. The light particles produced by the reaction were identified and measured with the *VAMOS++* magnetic spectrometer[13] placed at 0° with respect to the beam direction, using two drift chambers, a plastic scintillator and two small drift chambers placed at the entrance of the spectrometer. This led to an unambiguous identification of the $^{4}$He particles. The $^{4}$He particles were detected up to an angle in the laboratory of 10.0(5)° relative to the beam axis. The excitation energy ($E_x$), the velocity at the time of the reaction ($\beta_{\text{reac}} = v_{\text{reac}}/c$), and the angle of the $^{23}$Mg nuclei ($\theta_{\text{recoil}}$) have been determined by measuring the momentum of the $^{4}$He ejectiles, with a resolution (FWHM) of ≈500 keV, ≈0.0005 and ≈ 0.05°, respectively.

The excited states in $^{23}$Mg have been also clearly identified via their γ-ray transitions measured with the *AGATA* γ-ray spectrometer[14,15]. The present experimental method has taken advantage of the highest angle sensitivity of *AGATA*[14,15], shown in Fig. 1b. This γ-ray spectrometer is based on γ-ray tracking techniques in highly segmented HPGe detectors. It consisted of 31 crystals covering an angular range from 120° to 170°, with a total geometrical efficiency ≈ 0.6π. With *AGATA*, it is possible to measure accurately the energy and the emission angle of the γ rays with a resolution of 4.4(1) keV at 7 MeV and 0.7(1)°, respectively. The γ rays were observed Doppler shifted since they were emitted from a $^{23}$Mg nucleus moving at a certain velocity $\beta_{\text{ems}} = v_{\text{ems}}/c$. The measured energy $E_\gamma$ is a function of $\beta_{\text{ems}}$, of the center-of-mass energy $E_{\gamma,0}$ of the γ ray and of the angle $\theta$ between the γ ray and the $^{23}$Mg emitting nucleus: $E_\gamma = E_{\gamma,0}\sqrt{1 - \beta_{\text{ems}}^2}/(1 - \beta_{\text{ems}}\cos(\theta))$. With *AGATA*, this relativistic Doppler effect can be observed continuously as a function of the angle, as shown in Fig. 1a for the case of the $E_{\gamma,0} = 4840.0^{+0.2}_{-0.4}$ keV (we use energy values from the present measurement throughout this article) γ-ray transition emitted from the $E_x = 5292.0(6)$ keV excited state in $^{23}$Mg moving with $\beta_{\text{ems}} \approx 0.075$. Conversely, if the center-of-mass energy of the γ ray is known, it is possible to determine the nucleus velocity at the time of emission $\beta_{\text{ems}}$ from the measured $E_\gamma$ and $\theta$ (see the "Methods" subsection "Determination of velocities"). Their precise measurement, enabled by the combination of both *AGATA* and *VAMOS++*, has allowed for the accurate determination of $\beta_{\text{ems}}$, on an event-by-event basis.

In the present work, both velocities $\beta_{\text{ems}}$ and $\beta_{\text{reac}}$ were measured simultaneously, on an event-by-event basis. The two velocities are not

identical since (i) the $^{23}$Mg nucleus slows down in the target before being stopped (≈500 fs) and (ii) there is a time difference between the reaction and the γ-ray emission due to the finite lifetime of the state. Thus, the profile of the velocity differences, $\Delta\beta = \beta_{\text{reac}} - \beta_{\text{ems}}$, is a function of the lifetime of the state—a longer lifetime gives a larger value of $\Delta\beta$. The technique proposed here, for measuring femtosecond lifetimes, is based on the analysis of velocity-difference profiles.

The results obtained in this work are shown in Fig. 2 for three different excited states of $^{23}$Mg. These measurements were made using an average beam intensity of $2 \times 10^7$ pps for a duration of ~132 h. In Fig. 2a, $\beta_{\text{reac}}$ is shown as a function of $\beta_{\text{ems}}$, and in (b), the yields of the corresponding γ rays as a function of the velocity difference $\Delta\beta$. The overall shape of the $\Delta\beta$ measured profiles can be explained in a simple way. The γ rays are emitted following the exponential decay law, i.e. $N(t) = N_0 \times e^{-\Delta t/\tau}$, with $\tau$ being the lifetime of the state and $\Delta t$ the time elapsed after the reaction. For short lifetimes ($\tau \lesssim 100$ fs), the deceleration acting on the $^{23}$Mg ions is almost constant ($dE/dx \approx$ constant) and consequently $\Delta\beta \propto \Delta t$. Therefore, the right side tail of the curves in Fig. 2b would follow the exponential decay function $N(t) \propto e^{-\Delta\beta/\tau}$. This explains the asymmetric shape of the profile shown in blue, which corresponds to $\tau = 40$ fs. Moreover, the experiment has a resolution in $\Delta\beta$, i.e. 0.0032(1), which is measured by the width of the almost Gaussian profile observed for very short lifetimes, see the green profile for $\tau = 4$ fs in Fig. 2b. The overall shape of the measured $\Delta\beta$ distributions is the convolution of the exponential decay with a Gaussian function. The sensitivity of the method, governed by the statistics, can be better than the resolution of $\Delta\beta$. This makes the method sensitive to very short lifetimes, down to 0.8 fs according to simulations. From comparison of the data with Monte Carlo simulations, shown in Fig. 2b by the continuous lines (see also the Supplementary information), the best agreement is obtained with lifetimes of $4^{+1}_{-3}$ and $40^{+6}_{-7}$ fs for the states $E_x = 5292.0(6)$ keV ($E_{\gamma,0} = 4840.0^{+0.2}_{-0.4}$ keV) and $E_x = 3796.8(12)$ keV ($E_{\gamma,0} = 3344.8^{+0.8}_{-1.0}$ keV), respectively, which is in excellent agreement with their known values of 5(2)[16] and 41(6) fs[16]. This shows that the present method is a powerful tool to determine lifetimes in the femtosecond range. With the same method, it is also possible to accurately measure the center-of-mass energy of the γ-ray transitions. For the transition from the key state ($E_x = 7785.0(7)$ keV), we measured $E_{\gamma,0} = 7333.0^{+0.5}_{-0.2}$ keV, in good agreement with the referenced value of $E_{\gamma,0} = 7333.2(11)$ keV[16].

The present method has many advantages. Since the excitation energy of the state is selected with *VAMOS++*, we can ignore any possible top-feeding contribution to the state, and, therefore, the measure is not affected by the lifetime of higher-lying states. Moreover, this method is independent of the reaction mechanism

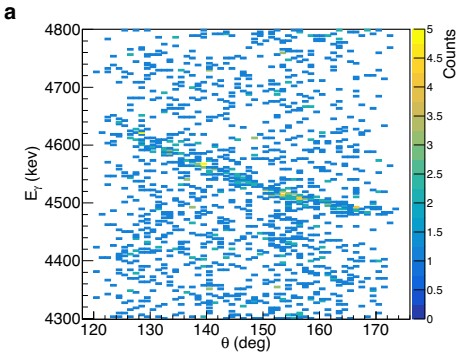

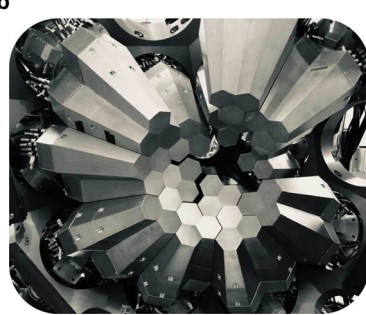

**Fig. 1 | Identification of γ-ray transitions in $^{23}$Mg. a** The energy of the measured γ rays is plotted as a function of the angle between the γ ray and the $^{23}$Mg emitter. This matrix is conditioned with the detection of an α particle at $5.2 < E_x < 5.4$ MeV in the *VAMOS++* magnetic spectrometer. The γ-ray transition ($E_{\gamma,0} = 4840.0^{+0.2}_{-0.4}$ keV) from the $E_x = 5292.0(6)$ keV excited state in $^{23}$Mg is clearly observed. Its energy is

Doppler shifted. The background observed here is mostly due to random coincidences between γ rays from the Compton background and α particles produced in fusion-evaporation reactions between the beam and $^{12}$C and $^{16}$O impurities deposited on the target. **b** Picture of the *AGATA* γ-ray spectrometer used to detect the γ rays emitted during the reaction.

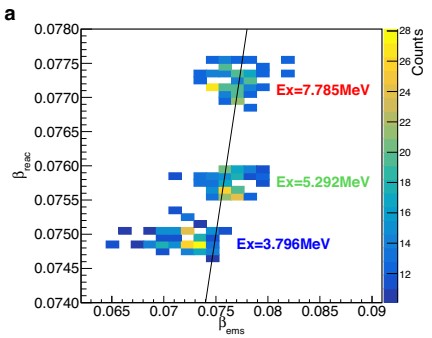
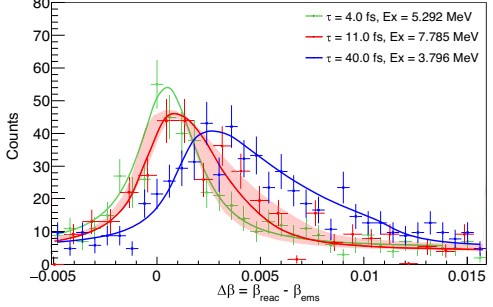

**Fig. 2 | Angle-integrated velocity-difference profiles. a** The $^{23}$Mg velocity at the time of reaction ($\beta_{reac}$) is shown against the velocity at the time of the γ-ray emission ($\beta_{ems}$), for three excited states. The line corresponds to the prompt γ-ray emission when $\beta_{ems} = \beta_{reac}$. The points observed on the left of the line ($\beta_{ems} < \beta_{reac}$) correspond to delayed γ-ray emissions. **b** The corresponding angle-integrated velocity-difference profiles for the three states compared with simulations (continuous lines). It shows unambiguously that the key state (in red) has a lifetime $4 < \tau < 40$ fs. The red-shaded area corresponds to the simulations with lifetimes within $1\sigma$ uncertainty. The horizontal error bars correspond to the width of the bins, which are larger than the real experimental uncertainty, and the vertical error bars to the statistical uncertainty.

populating the states, i.e. direct transfer or compound-nucleus formation ($^{24}$Mg + $^3$He → $^{27}$Si* → $^{23}$Mg + $^4$He). Furthermore, the result does not depend on the angular distribution of the emitted γ ray and of the charged particles, nor on the angle-dependent detection efficiency. A single spectrum concentrates all the statistics of the experiment, which maximizes the signal-to-noise ratio. The statistical uncertainty on the lifetime of the key state is ~50%. The main sources of systematic uncertainties are estimated to be 26% and the individual contributions are presented in Table 1.

The result obtained for the lifetime of the astrophysical state is $\tau = 11^{+7}_{-5}$ fs, including statistical and systematic uncertainties (see Fig. 2), confirming the measured value of Jenkins et al.[4] ($\tau = 10(3)$ fs). The reduced magnetic dipole transition probability $B(M1)$ was deduced from the measured lifetime assuming a negligible $E2$ electric quadrupole contribution: $B(M1) = 0.017^{+0.011}_{-0.008}\ \mu^2_N$. This value is among the lowest values measured for $M1$ transitions. To get insight from theory, we have performed shell-model calculations in the $sd$ shell with the well-established phenomenological USDA and USDB[17,18] interactions using the *NushellX@MSU* code[19]. These calculations confirm that the key state is characterized by a low $B(M1)$ value of 0.14 $\mu^2_N$, assuming 7/2$^+$ spin and parity assignment (see the discussion below) and optimum values of the effective proton and neutron $g$-factors. Although this value is almost a factor of 10 larger than the experimentally determined one, the difference is not far from the typical rms error on $M1$ transition probabilities established for the $sd$ shell with these interactions[11,20]. Therefore, the measured values of the lifetime are compatible with shell-model calculations.

The measured lifetime was used to determine the strength of the resonance: $\omega\gamma = \frac{2J_{^{23}Mg} + 1}{(2J_{^{22}Na} + 1)(2J_p + 1)} \times \frac{\Gamma_\gamma \Gamma_p}{\Gamma_\gamma + \Gamma_p} = \omega BR_p (1 - BR_p)\frac{\hbar}{\tau}$, with $J$ and $\Gamma$ being the spin and the partial widths of the key state. Actually, the spin of this state has also been a subject of debate[4,21], where three values have been proposed: $J^\pi_{^{23}Mg} = 3/2^{+\,21}$, $5/2^{+21}$ or $7/2^{+4}$. We assume it to be 7/2$^+$. This state is only 18(1.5) keV away from the known 5/2$^+$ isobaric analog state of the $^{23}$Al ground state. In the case of the 5/2$^+$ assignment, the states would interfere, as predicted by the nuclear shell-model calculations[11], and such mixing has not been observed[5,9,22,23]. Moreover, concerning the γ-ray decay pattern, shell-model calculations agree with a 7/2$^+$ spin[11]. The proton branching ratio, $BR_p$, was also measured in this work by detecting protons emitted from the $^{23}$Mg unbound states with an annular silicon detector placed downstream of the target (see Supplementary information). The obtained value, $BR_p = 0.68(17)\%$, is in excellent agreement with the latest published value, $BR_p = 0.65(8)\%$[23].

Combining all measured values into the new recommended values: $BR_p = 0.66(7)\%$ and $\tau = 10.2(26)$ fs, results in a consolidated resonance strength of $\omega\gamma = 0.24^{+0.11}_{-0.04}$ meV, which is compatible with the direct measurement[10] ($\omega\gamma < 0.36$ meV) but not with the other two direct measurements. It should be pointed out that these direct measurements using radioactive targets are not in agreement with each other, and also that the obtained value is very low, close to the sensitivity limit of these direct experiments. The new value of the resonance strength and a Monte-Carlo approach[24] were used to deduce a new rate for the $^{22}$Na($p, \gamma$)$^{23}$Mg reaction (see Fig. 4 and Table 1 in Supplementary information). The reliably estimated experimental uncertainties of the present method allowed a more accurate determination of this rate. This new rate was found to be very reliable at maximum nova temperatures, with uncertainties reduced to 40% (10%) at $T = 0.1$ GK (0.5 GK) (see Supplementary information).

To quantitatively assess the impact of the new $^{22}$Na($p, \gamma$)$^{23}$Mg reaction rate obtained from this work on nova nucleosynthesis, a series of hydrodynamic simulations have been performed. Four physical magnitudes determine the strength of a nova outburst, and, in turn, the synthesis of $^{22}$Na: the white dwarf mass $M_{WD}$ (or radius $R_{WD}$), its initial luminosity $L_{WD,ini}$, the mass-accretion rate $\dot{M}$, and the metallicity of the accreted material. The influence of the white dwarf mass on the synthesis of $^{22}$Na has been analyzed. Three different values for the white dwarf mass have been considered: 1.15 $M_\odot$, 1.25 $M_\odot$, and 1.35 $M_\odot$. In these simulations, the star hosting the nova outbursts is assumed to be an ONe white dwarf, with initial luminosity, $L_{WD,ini} = 10^{-2}\ L_\odot$, and

## Table 1 | Systematic uncertainties (in %) on the lifetime of the key state

| Source of uncertainty | Uncertainty on $\tau$ in % |
|---|---|
| γ-ray energy resolution[a] | 15 |
| γ-ray absolute angle (1°)[b] | 12 |
| γ-ray angular resolution[a] | 12 |
| Stopping powers[c] | 10 |
| Beam energy dispersion[a] | 6.5 |
| Implementation profile of $^3$He ions[c] | 6.0 |
| γ-ray energy shift during runs[d] | 3.5 |
| Transverse spatial dispersion of the beam[a] | 0.3 |
| Total | 26 |

[a]Measured during the experiment.
[b]Including the uncertainty in the position of the target.
[c]From *SRIM*[39]-*EVASIONS*[40] simulations, considering 20% uncertainty in stopping powers tables.
[d]From the energy calibration stability measured throughout the experiment.

**Table 2 | Abundances of (Ne, Na) isotopes in the ejecta obtained for different nova models calculated with the *MESA* and *SHIVA* codes (see text for the parameters)**

| Model | 115a | 115b | 125 | 135 |
|---|---|---|---|---|
| HD code | *MESA* | *SHIVA* | *SHIVA* | *SHIVA* |
| $M_{WD}$ ($M_\odot$) | 1.15 | 1.15 | 1.25 | 1.35 |
| $R_{WD}$ (km) | 4428 | 4334 | 3797 | 2258 |
| $T_{peak}$ ($10^8$ K) | 2.12 | 2.27 | 2.48 | 3.13 |
| $M_{ejec}$ ($10^{-5} M_\odot$) | 4.63 | 2.46 | 1.90 | 0.46 |
| X($^{20}$Ne) | $1.4 \times 10^{-1}$ | $1.8 \times 10^{-1}$ | $1.8 \times 10^{-1}$ | $1.5 \times 10^{-1}$ |
| X($^{21}$Ne) | $2.3 \times 10^{-5}$ | $3.6 \times 10^{-5}$ | $3.9 \times 10^{-5}$ | $4.1 \times 10^{-5}$ |
| X($^{22}$Ne) | $1.9 \times 10^{-3}$ | $1.3 \times 10^{-3}$ | $6.4 \times 10^{-4}$ | $3.3 \times 10^{-5}$ |
| **X($^{22}$Na)** | **$3.1 \times 10^{-4}$** | **$3.2 \times 10^{-4}$** | **$3.7 \times 10^{-4}$** | **$9.1 \times 10^{-4}$** |
| X($^{23}$Na) | $8.2 \times 10^{-4}$ | $9.9 \times 10^{-4}$ | $1.1 \times 10^{-3}$ | $3.4 \times 10^{-3}$ |

Bold entries highlight the abundance results of the radioisotope $^{22}$Na.

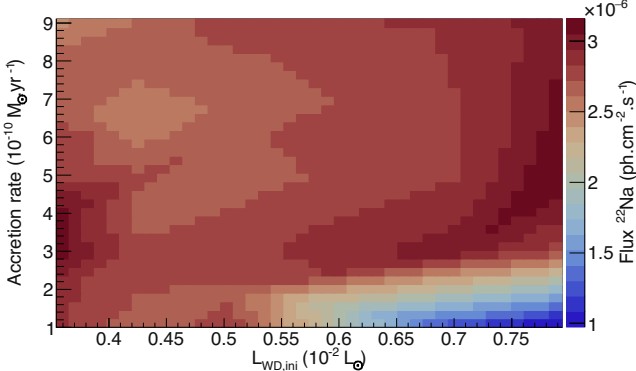

**Fig. 3 | Prediction of the 1.275 MeV γ-ray flux emitted from a nova.** The $^{22}$Na γ-ray emission flux is shown as a function of the white dwarf initial luminosity and the mass-accretion rate. This is calculated from the $^{22}$Na mass-averaged abundance within the ejected shells. Computations were done for a 1.2 $M_\odot$ ONe white dwarf located 1 kpc from the Earth, using the *MESA* code[27–32].

accreting solar composition material[25] from the secondary star at a constant mass-accretion rate of $\dot{M} = 2 \times 10^{-10} M_\odot$/yr. The accreted matter is assumed to mix with material from the outermost white dwarf layers[26]. For consistency and completeness, the nova outbursts on 1.15 $M_\odot$ white dwarfs have been computed with two different, one-dimensional stellar evolution codes: *MESA*[27–32] and *SHIVA*[33,34].

At the early stages of the outburst, when the temperature at the base of the envelope reaches $T_{base} \sim 5 \times 10^7$ K, the chain of reactions $^{20}$Ne$(p,\gamma)^{21}$Na$(\beta^+)^{21}$Ne$(p,\gamma)^{22}$Na powers a rapid rise in the $^{22}$Na abundance. When the temperature reaches $T_{base} \sim 8 \times 10^7$ K, $^{22}$Na$(p,\gamma)^{23}$Mg becomes the main destruction channel. At $T_{base} \sim 10^8$ K, $^{22}$Na$(p,\gamma)^{23}$Mg becomes the most important reaction involved in the synthesis and destruction of $^{22}$Na, and therefore, its abundance begins to decrease. When $T_{base} \sim 2 \times 10^8$ K, $^{21}$Na$(p,\gamma)^{22}$Mg becomes faster than $^{21}$Na$(\beta^+)^{21}$Ne, such that $^{20}$Ne$(p,\gamma)^{21}$Na$(p,\gamma)^{22}$Mg$(\beta^+)^{22}$Na takes over as the dominant path, favoring the synthesis of $^{22}$Na, which achieves a second maximum after the temperature peak $T_{peak}$ (see Table 2). This pattern continues up to $^{22}$Na$(p,\gamma)^{23}$Mg becomes again the most relevant reaction when the temperature drops. During the subsequent and final expansion and ejection stages, as the temperature drops dramatically, the evolution of $^{22}$Na is fully governed by $^{22}$Na$(\beta^+)^{22}$Ne. The most relevant results obtained in these simulations are summarized in Table 2. The larger peak temperatures achieved during nova outbursts on more massive white dwarfs yield larger mean, mass-averaged abundances of $^{22}$Na in the ejecta. However, it is worth noting that the total mass of $^{22}$Na ejected in a nova outburst decreases with the white dwarf mass ($8 \times 10^{-9} M_\odot$, for Model 115b, $7 \times 10^{-9} M_\odot$, for Model 125, and $4.2 \times 10^{-9} M_\odot$, for Model 135), since more massive white dwarfs accrete and eject smaller amounts of mass, see $M_{ejec}$ in Table 2.

The new and reliable reaction rate obtained in this work also opens the door to further advanced sensitivity studies on astrophysical parameters, performed here for the first time. The flux has been obtained in a series of nova simulations for a large range of mass-accretion rates $\dot{M}$ and white dwarf luminosities $L_{WD,ini}$. For example, Fig. 3 shows the expected flux of $^{22}$Na for a nova event located 1 kpc from the Earth, with $M_{WD} = 1.2 M_\odot$ and accreting solar composition material. This figure shows that the amount of $^{22}$Na does depend on the two parameters, varying smoothly by up to a factor of 3. The change is abrupt only on the lower right side of the figure. In this region, the $^{22}$Na/$^{22}$Mg and $^{22}$Na/$^{21}$Ne ratios around the peak temperatures have the same values as in the other regions, which means that the production pathways are unchanged. On the contrary, the $^{23}$Mg/$^{22}$Na ratio is found

4 times larger, indicating that the destruction by the $^{22}$Na$(p,\gamma)^{23}$Mg reaction is more intense in this region. This graph provides a link between the unknown astrophysical parameters of the novae and the predicted $^{22}$Na flux. In the future, precise measurements of the $^{22}$Na γ-ray flux will constrain these astrophysical parameters.

New instruments for γ-ray astronomy are under study or under construction: ESA's enhanced e-ASTROGAM[35] and NASA's COmpton Spectrometer and Imager, COSI[36]. Both are, in principle, capable of detecting the 1.275 MeV γ rays released in the decay of $^{22}$Na produced in nova outbursts, since they are being designed with a higher efficiency than the past missions, INTEGRAL-SPI[37] and COMPTEL-CGRO[38]. The very high precision in the determination of the $^{22}$Na$(p,\gamma)^{23}$Mg rate reported in this work permits deriving, for the first time, realistic estimates of the maximum detectability distance of novae in γ rays, through the 1.275 MeV line. With the expected sensitivities of e-ASTROGRAM ($3 \times 10^{-6}$ photons cm$^{-2}$ s$^{-1}$) and COSI ($1.7 \times 10^{-6}$ photons cm$^{-2}$ s$^{-1}$) at 1 MeV, and assuming model 125 (Table 2), the new maximum detectability distances of 2.7 and 4.0 kpc, respectively, have been derived, on which the reaction rate uncertainty obtained here leads to an uncertainty of 18%. These new estimates suggest a large chance for the possible detection of the $^{22}$Na γ rays produced in ONe novae by the next generation of space-borne γ-ray observatories.

## Methods
### Determination of velocities
The velocity $\beta_{reac}$ of the $^{23}$Mg nuclei at the reaction time was derived from the momentum of the α particles measured with the *VAMOS++* magnetic spectrometer and from the kinematics laws of energy and momentum conservation in the case of the $^3$He($^{24}$Mg,$\alpha$)$^{23}$Mg two-body reaction

$$\beta_{reac} = \sqrt{1 - \gamma_{reac}^{-2}} \tag{1}$$

with

$$\gamma_{reac} = \sqrt{1 + \frac{\frac{m_\alpha^2 \beta_\alpha^2}{1-\beta_\alpha^2} + \frac{m_{beam}^2 \beta_{beam}^2}{1-\beta_{beam}^2} - 2\cos(\theta_\alpha) m_{beam} m_\alpha \sqrt{\gamma_\alpha^2 - 1}\sqrt{\gamma_{beam}^2 - 1}}{m_{recoil}^2}} \tag{2}$$

where $m_\alpha$, $m_{beam}$ and $m_{recoil}$ are the rest-mass energies of the $\alpha$, $^{24}$Mg, and $^{23}$Mg nuclei, $\theta_\alpha$ the angle between the $\alpha$ particle and the beam axis. The parameter $\gamma_\alpha$ was measured with *VAMOS*++ and corrected for the energy losses in the target using the *SRIM* code[39]. The parameter $\gamma_{beam}$ was measured prior to the experiment, $\gamma_{reac}$ was then determined from the measured $\gamma_\alpha$ and a $\gamma$-ray transition was detected in coincidence.

The velocity $\beta_{ems}$ of the $^{23}$Mg nuclei at the $\gamma$-ray emission time was derived from the measured $\gamma$ rays. Since the $^{23}$Mg nuclei are moving at the time of the $\gamma$-ray emission, the $\gamma$-ray energy is Doppler shifted, with the measured energies $E_\gamma$ shifted from the center-of-mass energy $E_{\gamma,0}$ according to

$$E_\gamma = E_{\gamma,0} \frac{\sqrt{1 - \beta_{ems}^2}}{1 - \beta_{ems}\cos(\theta)} \quad (3)$$

It follows that

$$\beta_{ems} = \frac{R^2\cos(\theta) + \sqrt{1 + R^2\cos^2(\theta) - R^2}}{R^2\cos^2(\theta) + 1} \quad (4)$$

with $R = E_\gamma/E_{\gamma,0}$. Here, $R < 1$ since the *AGATA* detector was located upstream of the target. The angle $\theta$ between the $\gamma$ ray and the $^{23}$Mg recoil nucleus was derived from the measured $(\theta, \phi)$ of the $\gamma$ ray and the $\alpha$ particle using the formulas

$$\cos(\theta) = \sin(\theta_\gamma)\sin(\theta_{recoil})[\cos(\phi_\gamma)\cos(\phi_{recoil})$$
$$+ \sin(\phi_\gamma)\sin(\phi_{recoil})] + \cos(\theta_\gamma)\cos(\theta_{recoil})$$

where

$$\theta_{recoil} = \mathrm{acos}\left(\frac{m_{beam}\sqrt{\gamma_{beam}^2 - 1} - m_\alpha\cos(\theta_\alpha)\sqrt{\gamma_\alpha^2 - 1}}{m_{recoil}\sqrt{\gamma_{recoil}^2 - 1}}\right) \text{ and } \phi_{recoil} = \pi + \phi_\alpha$$

### Fit of velocity-difference profiles
Velocity-difference profiles were numerically simulated with a Monte-Carlo approach developed in the *EVASIONS* C++/ROOT code[40]. Simulated velocity-difference profiles were normalized to the measured ones via the profile integrals. The goodness of fit between experimental and simulated profiles was quantified with the Pearson $\chi^2$ tests where the lifetime and the $\gamma$-ray center-of-mass energy were taken as free parameters.

### Branching ratios
The $E_x = 7785.0(7)$ keV astrophysical state can decay via proton or $\gamma$-ray emission. Therefore, after applying a selection on $E_x$ in $^{23}$Mg by using the measured $\alpha$ particles, the number of detected protons and $\gamma$ rays allowed us to determine the proton and $\gamma$-ray branching ratios. These values were corrected for detection efficiencies. On the one hand, the geometrical efficiency of the silicon detector was estimated by numerical simulations. The angular distribution was considered isotropic for the emitted $\ell = 0$ protons. On the other hand, the *AGATA* efficiency was measured at low energies with a radioactive $^{152}$Eu source, and simulated with the *AGATA Geant*4 code library[41,42] to determine the efficiency at high energies after scaling the simulations to the measured efficiencies at low energies.

### Determination of the $^{22}$Na flux in novae
The amount of $^{22}$Na ejected in a nova outburst was obtained with the simulation codes *MESA*[27–31] and *SHIVA*[33,34] from its abundance in the ejected layers. *SHIVA* and *MESA* use two basic criteria for the ejection of a specific layer: if its velocity achieves the escape velocity (~1000 km s$^{-1}$), or if its luminosity becomes higher than the Eddington

limit (when the force exerted by radiation exceeds the gravitational pull) and hence, the condition of hydrostatic equilibrium no longer holds.

## Data availability
The data used in this study originate from the E710 GANIL dataset[43]. The ownership of data generated by the *AGATA* $\gamma$-ray spectrometer resides with the *AGATA* collaboration as detailed in the *AGATA* Data Policy[44]. The source data of the figures[45] are provided in the Supplementary information/Source Data file.

## Code availability
The *EVASIONS* code[40], used in this study to analyze the experimental data, is available at https://github.com/CFougeres/EVASIONScode. The code is briefly described in Supplementary information. Other codes employed here, i.e. *SRIM*[39], *NushellX@MSU*[19], *MESA*[27–31] and *RatesMC*[24], are freely available.

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

## Acknowledgements

The authors thank the GANIL accelerator staff for their beam delivery and support as well as the *AGATA* collaboration. This work was supported by the Normandie Region, the European project ChETEC-INFRA (101008324), ASTRANUCAP IRN, NuAG IRP, SPIRAL2-CZ, and the U.S. Department of Energy, Office of Science, Office of Nuclear Physics, under contract number DE-AC02-06CH11357. This work was also partially supported by the MINECO grants PID2020-117252GB-I00, PID2020-118265GB-C41 and PID2020-118265GB-C44, and by the AGAUR/Generalitat de Catalunya grant SGR-386/2021. V.G. would like to thank São Paulo Research Foundation (FAPESP) (grant 2016/02863-4) and CNPq (grant 302969/2013-6). Y.H.K. would like to acknowledge the support of the Institute for Basic Science (IBS-R031-D1). M.S. would like to acknowledge that this work has been partially supported by the OASIS project no. ANR-17-CE31-0026.

## Author contributions

F.O.S. and C.M. were PIs of the experiment during which F.O.S., C.M., E.C., Y.H.K., A.L., V. Guimarães, D. Bemmerer, F.B., I.C., M.C., C.D., J.D., Z.F., S.L., J.L., A.L.-M., J.M., D.R., N.R., A.M.S.-B, M. Stanoiu, and P.U. participated in person. Targets making was performed by R.B. C.F., C.M., E.C., A.L., D. Barrientos, G.B., A.J.B., A.B., B.C., C.D.-P., J.D., J.E., V. González, A.G, J.G., H.H., A.J., A. Kaşkaş, A. Korichi, S.M.L., S.L., H.L., J.L., A.L.-M., R.M., D.M., B.M., D.R.N., J.N., Zs.P., A.P., B.Q., D.R., P.R., K.R., F.S., M.D.S., E.S., M. Şenyiğ, M. Siciliano, D.S., Ch.T., J.J.V.-D., and M.Z. have been involved to the *AGATA* collaboration. Experimental analysis and simulations were undertaken by C.F. and F.O.S. Experimental results were in particular discussed by C.F., F.O.S., E.C., A.L., and A.N. Theoretical calculations on nuclear structure were done by C.F., N.S., and F.O.S. Astrophysical calculations and analysis were performed by C.F., J.J. and F.O.S. F.O.S., C.F., and J.J. were involved in the primary manuscript. All co-authors discussed the final results and contributed to the final manuscript.

## Competing interests

The authors declare no competing interests.

## Additional information

Chloé Fougères [1,2] ✉, François de Oliveira Santos [1] ✉, Jordi José [3,4], Caterina Michelagnoli [1,5], Emmanuel Clément[1], Yung Hee Kim [1,6], Antoine Lemasson [1], Valdir Guimarães [7], Diego Barrientos [8], Daniel Bemmerer [9], Giovanna Benzoni[10], Andrew J. Boston[11], Roman Böttger[9], Florent Boulay[1], Angela Bracco[10], Igor Čeliković [12], Bo Cederwall [13], Michał Ciemala [14], Clément Delafosse [15], César Domingo-Pardo[16], Jérémie Dudouet [17], Jürgen Eberth[18], Zsolt Fülöp [19], Vicente González [20], Andrea Gottardo [21], Johan Goupil[1], Herbert Hess[18], Andrea Jungclaus[22], Ayşe Kaşkaş [23], Amel Korichi[15], Silvia M. Lenzi [24,25], Silvia Leoni [10,26], Hongjie Li[1], Joa Ljungvall[15], Araceli Lopez-Martens [15], Roberto Menegazzo [24], Daniele Mengoni[24,25], Benedicte Million [10], Jaromír Mrázek [27], Daniel R. Napoli[21], Alahari Navin[1], Johan Nyberg [28], Zsolt Podolyák [29], Alberto Pullia [10,26], Begoña Quintana [30], Damien Ralet [1,15], Nadine Redon[17], Peter Reiter [18], Kseniia Rezynkina[24,31], Frédéric Saillant[1], Marie-Delphine Salsac[32], Angel M. Sánchez-Benítez[33], Enrique Sanchis [20], Menekşe Şenyiğit [23], Marco Siciliano [2,32], Nadezda A. Smirnova[34], Dorottya Sohler [19], Mihai Stanoiu [35], Christophe Theisen [32], Jose J. Valiente-Dobón[21], Predrag Ujić[12] & Magdalena Zielińska[32]

[1]Grand Accélérateur National d'Ions Lourds (GANIL), CEA/DRF-CNRS/IN2P3, Caen, France. [2]Physics Division, Argonne National Laboratory, Lemont, IL 60439, USA. [3]Departament de Física, EEBE, Universitat Politècnica de Catalunya, Barcelona, Spain. [4]Institut d'Estudis Espacials de Catalunya, Barcelona, Spain. [5]Institut Laue-Langevin, Grenoble, France. [6]Center for Exotic Nuclear Studies, Institute for Basic Science, Daejeon, Republic of Korea. [7]Instituto de Física, Universidade de São Paulo, CEP 05508-090 São Paulo, Brazil. [8]CERN, Geneva, Switzerland. [9]Helmholtz-Zentrum Dresden-Rossendorf, Institute of Ion Beam Physics and Materials Research, Dresden, Germany. [10]Istituto Nazionale di Fisica Nucleare, Sezione di Milano, Milano, Italy. [11]Oliver Lodge Laboratory, University of Liverpool, Liverpool, UK. [12]Vinca Institute of Nuclear Sciences, University of Belgrade, Belgrade, Serbia. [13]Department of Physics, KTH Royal Institute of Technology, Stockholm, Sweden. [14]Institute of Nuclear Physics Polish Academy of Sciences, PL-31342 Kraków, Poland. [15]Laboratoire de Physique des 2 Infinis Irène Joliot-Curie, CNRS/IN2P3, Université Paris-Saclay, Orsay, France. [16]Instituto de Física Corpuscular, CSIC-Universidad de Valencia, Valencia, Spain. [17]Université Lyon, Université Claude Bernard Lyon 1, CNRS/IN2P3, IP2I Lyon, UMR 5822, F-69622 Villeurbanne, France. [18]Institut für Kernphysik, Universität zu Köln, Köln, Germany. [19]Institute for Nuclear Research (ATOMKI), Debrecen, Hungary. [20]Departamento de Ingeniería Electrónica, Universitat de Valencia, Valencia, Spain. [21]Laboratori Nazionali di Legnaro INFN, Legnaro, Italy. [22]Instituto de Estructura de la Materia, CSIC, Madrid, Spain. [23]Department of Physics, Faculty of Science, Ankara University, Ankara, Turkey. [24]Istituto Nazionale di Fisica Nucleare, Sezione di Padova, Padova, Italy. [25]Dipartimento di Fisica e Astronomia, Universitá degli Studi di Padova, Padova, Italy. [26]Dipartimento di Fisica, Università di Milano, Milano, Italy. [27]Nuclear Physics Institute of the Czech Academy of Sciences, Řež, Czech Republic. [28]Department of Physics and Astronomy, Uppsala University, Uppsala, Sweden. [29]Department of Physics, University of Surrey, Guildford, UK. [30]Laboratorio de Radiaciones Ionizantes, Departamento de Física Fundamental, Universidad de Salamanca, Salamanca, Spain. [31]Institut pluridisciplinaire Hubert Curien, Université de Strasbourg, CNRS, Strasbourg, France. [32]Irfu, CEA, Université Paris-Saclay, Gif-sur-Yvette, France. [33]Department of Integrated Sciences, Centro de Estudios Avanzados en Fisica, Matematicas y Computacion (CEAFMC), University of Huelva, Huelva, Spain. [34]Université de Bordeaux, CNRS/IN2P3, LP2IB, Gradignan, France. [35]Horia Hulubei National Institute for R&D in Physics and Nuclear Engineering, IFIN-HH Bucharest, Măgurele, Romania. ✉e-mail: chloe.fougeres@gmail.com; francois.oliveira@ganil.fr

