## [Peer Review File · Nature Communications]

Search for ^{22}Na in novae supported by a novel method for measuring femtosecond nuclear lifetimesREVIEWER COMMENTS

Reviewer #1 (Remarks to the Author):

The authors reported the lifetime measurements of the excited states of ^{22}Na by a novel method for astrophysical interests. The subject of this work is important and has been investigated as a topic in the field of astrophysics. Experiment is challenging to study the lifetime of the astrophysical state of ^{22}Na . The lifetime and the proton branching ratio of the state measured in this work are in good agreement with the published values. Additionally, the new recommended values of the lifetime and the proton branching ratio were used to estimate the $^{22}\text{Na}(p,\gamma)^{23}\text{Mg}$ rate by hydrodynamic simulations. The estimations provide the constrained astrophysical rate compared with the published studies.

Femtosecond nuclear lifetimes of the excited states of ^{22}Na , measured by a combined analysis of particle-particle correlations and velocity-difference profiles in this work, are verified by comparing with the published values for the states for $E_x = 5292$ keV and 3796 keV. Thereafter, the lifetime of the 7785 keV state was determined to be 11^{+7}_{-5} fs. Its error is larger than the published value of Jenkins et al (10(3) fs) due to its low statistics. This analytical approach by comparing simulations is properly validated. The sensitivity of the approach is governed with the statistics. However, information on a ^{24}Mg beam, namely intensity, irradiation duration, and total beam dose, which is related to the statistics, is not present in the text. This must be improved.

The other experimental data obtained in this work, the proton branching ratio 0.68(6)%, is in excellent agreement with the published value 0.65(8)%. The experimental ratios are used to estimate the astrophysical rate for $^{22}\text{Na}(p,\gamma)^{23}\text{Mg}$. However, this paper lacks information on the measurement of protons emitted from the Mg unbound states with an annular silicon detector placed downstream of the target. This must be improved so that the experiments would be reproduced.

On the assumption of the spin of the state, combining all measured values in the past and present studies for the proton branching ratio and the lifetime, results in a resonance strength of $0.24^{+0.11}_{-0.04}$ meV. This provides the new $^{22}\text{Na}(p,\gamma)^{23}\text{Mg}$ rate by hydrodynamic simulations. The resonance strength of 0.24 meV is not in agreement with those of direct measurements, as pointed out in the text, but is in excellent agreement with the published value 0.24(8) meV compared with >0.16 meV of Ref. 27. Thus, this work constrains limits of the astrophysical rate, besides this would be helpful for planned balloon-borne experiments. The rate was re-evaluated by using the Monte Carlo statistical approach of the published studies, as shown in supplementary information. Thus, this approach is evaluated to be valid.

The present novel method would provide more strong impact when advantages discussed in the last paragraph of the text are experimentally confirmed.

Reviewer #2 (Remarks to the Author):

- What are the noteworthy results?

Certainly. The reaction is essential for the production of one of the principal signatures of explosive nucleosynthesis in novae. The community has been debating the possible measurement of the ^{22}Na decay line for decades without consensus.

- Will the work be of significance to the field and related fields? How does it compare to the established literature? If the work is not original, please provide relevant references.

No question. This is a major advance.

- Does the work support the conclusions and claims, or is additional evidence needed?

- Are there any flaws in the data analysis, interpretation and conclusions? - Do these prohibit publication or require revision?

See comments, the description of the experiments and related structure computations is excellent, with only a few, simply corrected, lacunae.

- Is the methodology sound? Does the work meet the expected standards in your field?

Yes.

- Is there enough detail provided in the methods for the work to be reproduced?

Yes.

Comments, all minor (a couple aren't merely editorial) but I hope useful:

1. Please identify the color bar in the caption.

2. In the discussion following Fig. 2, it would be useful to separate the statistical and systematic uncertainties.

3. line 257: "It is worth mentioning that ..."; I suggest providing the reaction rates in the usual temperature dependent form.

4. line 312: "and the measurable", it hasn't been measured yet. One more comment on this point: there are also reasons for thinking that the opacity of the ejecta might mitigate this predicted measurement so that might be added as a caveat, lest the casual reader think this is a firm observable. This also applies to the discussion on line 325ff.

5. line 320: "permits deriving, for the first time"

6. The material in the paragraph beginning at line 330 might fit better in a supplementary paragraph, I think it weakens the principal message of the paper.

7. line 368: It was annoying to read that the details behind the numerical simulation are unavailable, as a referee (and reader) I would have wanted to know these details, or at least more than the squib included in the text.

8. line 386: more detail is needed than "several criteria"

Reviewer #3 (Remarks to the Author):

This is a review of the manuscript entitled “Search for ^{22}Na in novae supported by a novel method for measuring femtosecond nuclear lifetimes” by C. Fougères et al. submitted to Nature Communications. The manuscript describes a new measurement of the lifetime of the 7785-keV state in ^{23}Mg , which corresponds to the $^{22}\text{Na} + p$ resonance that dominates the $^{22}\text{Na}(p,g)^{23}\text{Mg}$ reaction rate in classical nova nucleosynthesis. This reaction is the primary destruction mechanism for ^{22}Na , a gamma-ray emitting nucleus that has long been postulated to be observable in the Galaxy, but is yet undetected by gamma-ray satellite observatories. Due to the sensitivity of the ^{22}Na abundance on this reaction rate, a precise rate would allow more accurate predictions of the probability of this gamma-ray detection for future observations. This study describes a novel technique to measure lifetimes down to sub-fs precision, which has been used to measure this state’s lifetime and confirms the previous measurement. The weighted average of these measurements is then used to determine the reaction rate and predict the flux that may be observed by future planned gamma-ray missions.

The data given and the results and conclusions regarding the ^{22}Na gamma-ray flux derived from that data appear to be both valid and relatively robust. While the variation in WD mass, accretion rate, and WD luminosity give the reader an understanding of the uncertainty in the expected ^{22}Na flux due to the stellar parameters, that uncertainty due to the reaction rate uncertainty is not given, which would be useful in ascertaining the relative effects of these two types of uncertainties on the flux. Presumably both of these will affect the uncertainty in the final determination of the maximum detectability distances for future gamma-ray missions, which should be included (or at least the contribution of the reaction rate uncertainty given the focus of this manuscript on that new determination).

While the study combines a novel method with an in-depth analysis for the predicted gamma-ray flux from ^{22}Na , ultimately the measured lifetime is in agreement, albeit less precise, than that achieved by D. Jenkins et al. [11] and the resulting resonance strength is not significantly different from, e.g., Friedman et al., [27] when using the Jenkins et al. lifetime. Compared to previous studies, the stellar modeling using these results is more robust, using multiple CO WD masses, two different codes, etc.; however, the mass fraction of ^{22}Na determined in this study is relatively close the previous determinations of [11] and [27] with differences of approximately 10%. However, one can certainly be more confident in this new determination with the confirmation of the key state’s lifetime measured determined herein. Thus, the main significance of the study lies in the novel method used to measure lifetimes and the wide range of parameters used in the stellar modeling.

While gamma-spectroscopy is outside my area of expertise, in general the methodology seems sound, and the experimental equipment used is well understood. Perhaps the most uncertain part of the analysis lies in the simulations of the target profile and the stopping power of the ^{23}Mg in the Au foil. Tabulated stopping powers are well-known to be somewhat imprecise; however, the authors have used relatively conservative uncertainties of 20% for these tables in their error budget, which is the norm. In

the supplementary information, it is stated that these simulations/analysis tools have been validated by comparisons to known states. Providing more information here (which states were used if validation was done as part of this study, or a reference if one exists) would give the reader more confidence in these results.

In addition to the comments provided above, the following edits would improve the clarity and accuracy of the manuscript:

- The value quoted for the resonance strength determined by Ref. [13] in the manuscript is given as 1.4(3) meV; however, Stegmüller et al., give that value for the contribution to the resonance strength for the specific transition to the 451-keV state. Admittedly, this is given as a 100% branch by Jenkins et al., but nevertheless their recommended value for the resonance strength and the value used in their reaction rate calculations is given in their Table 1 as 1.8(7) meV. For consistency of comparing reaction rate results, the authors may wish to use this latter value.
- Reference [6] is given for the statement that the ejected mass of ^{22}Na in novae has been shown to be inversely proportional to the lifetime of the 7785-keV state; however, there is no discussion of the lifetime of this state in that publication.
- In lines 160 and 200 “rest energy” of the gamma ray is presumably referring to the center of mass energy? If so, using the latter terminology might help alleviate confusion for a more junior reader who may incorrectly interpret this as meaning rest mass.
- Is the shaded red region shown in Fig. 2 (right) the 1-sigma uncertainty of the simulation or the data? The caption should be amended to make this clear to the reader.
- Is there a source that can be cited for the statement that for short lifetimes the deceleration force is constant leading to a $\Delta\beta$ that is proportional to Δt ? Why this should be so is not immediately clear from the text.
- It seems that a temperature value may be missing in line 286.
- While the angular distributions should not affect the lifetime determination as stated in the paper, those likely should be included in the determination of the branching ratio. It is not clear from that section of the Methods part of the manuscript if this was taken into account.
- The in the supplemental material the authors compare their new reaction rate to those of Sallaska et al. and Stegmüller et al., and give the uncertainty of their newly determined rate. Including both the uncertainties in these previous determinations, as well as a comparison between this result and the recent reaction rates from Jenkins et al. and Friedman et al. would provide a more complete picture of the improvements due to this study compared to the literature.

Point-by-point response to the reviewers' comments, reproduced verbatim

Please find the responses to the referees' comments and queries.

Reviewer #1

i) However, information on a ^{24}Mg beam, namely intensity, irradiation duration, and total beam dose, which is related to the statistics, is not present in the text. This must be improved.

This following text has now been added.

“These measurements were made using an average beam intensity of 2×10^7 pps for a duration of ~132 hours. “

ii) The other experimental data obtained in this work, the proton branching ratio 0.68(6)%, is in excellent agreement with the published value 0.65(8)%. The experimental ratios are used to estimate the astrophysical rate for $^{22}\text{Na}(p,\gamma)^{23}\text{Mg}$. However, this paper lacks information on the measurement of protons emitted from the Mg unbound states with an annular silicon detector placed downstream of the target. This must be improved so that the experiments would be reproduced.

These aspects and the results for the proton branching ratio have been added in Supplementary information as a new section.

It should be noted that there was an error in the announced uncertainty. We had noted $\text{BR}_p=0.68(6)\%$, this has now changed to $\text{BR}_p=0.68(17)\%$. This has no consequence on the results and conclusions of the article.

iii) The present novel method would provide more strong impact when advantages discussed in the last paragraph of the text are experimentally confirmed.

Following the suggestions of reviewer #2, this last paragraph has been moved to the Supplementary information.

Reviewer #2

Comments, all minor (a couple aren't merely editorial) but I hope useful:

1. Please identify the color bar in the caption.

We have put the units (counts) in the revised version.

2. In the discussion following Fig. 2, it would be useful to separate the statistical and systematic uncertainties.

This following text has now been added: "The statistical uncertainty on the lifetime of the key state is ~50%. The main sources of systematic uncertainties are estimated to be 26% and the individual contributions are presented in Table 1."

3. line 257: "It is worth mentioning that ..."; I suggest providing the reaction rates in the usual temperature dependent form.

To avoid any possible confusion, we have removed the following sentence: "It is worth mentioning that if we consider the two possible assignments $J=(5/2+,7/2+)$, only a limited increase in the uncertainties is obtained, at most 11% below 0.25 GK". There is no reason to talk about 5/2+ spin, and an additional table on the reaction rate with 5/2+ may confuse the reader.

4. line 312: "and the measured", it hasn't been measured yet.

The word "measured" has been changed to "predicted"

One more comment on this point: there are also reasons for thinking that the opacity of the ejecta might mitigate this predicted measurement so that might be added as a caveat, lest the casual reader think this is a firm observable. This also applies to the discussion on line 325ff.

The effect of the opacity has been discussed in detail in a previous work (see, e.g., Gomez-Gomar et al. 1998, MNRAS, vol. 296, 913-920). The transition from optically-thick to optically-thin conditions in the ejecta (i.e., transparency to γ rays) of a classical nova occurs about a week after the explosion. This will certainly influence the γ -ray emission of short-lived signals (e.g., the 20-511 keV continuum arising mainly from electron-positron annihilation plus Comptonization, driven by the decay of ^{18}F), however for the case of the longer-lived 1275 keV γ -ray line associated to ^{22}Na decay, this will only mildly affect the rise of the light curve up to peak value (the rising phase of the 1275 keV γ rays-ray lasts between 10 and 20 days, depending of the model). As shown in Fig. 7 of the above mentioned Gomez-Gomar et al, the 1275 keV γ -ray line will remain with a flux comparable to the maximum value for a relatively long time. After a week, the opacity will no longer interfere with the emission of the 1275 keV γ -ray line, a signal potentially observable for about 3 years.

The above point strengthening ^{22}Na as a key candidate for observing classical novae at low-energy γ rays has been added in the introduction:

"With half-life of ^{22}Na being much longer than the transition time from an optically-thick to an optically-thin ejecta of a classical nova (about one week), the flux of the 1.275 MeV the 1.275 MeV γ -ray line will remain close to its maximum value for months after the ejecta has become transparent to γ rays."

5. line 320: "permits deriving, for the first time"

This has been corrected.

6. The material in the paragraph beginning at line 330 might fit better in a supplementary paragraph, I think it weakens the principal message of the paper.

As suggested, this has been moved to the Supplementary information.

7. line 368: It was annoying to read that the details behind the numerical simulation are unavailable, as a referee (and reader) I would have wanted to know these details, or at least more than the squib included in the text.

The code is briefly described in the *Numerical Simulations* section in the Supplementary information. The following sentence has been removed: "The principle of these numerical simulations will be the subject of a future publication". It has been replaced by a direct link to the code, in the *Code Availability* section: "The EVASIONS code, used in this study to analyze the experimental data, is available at <https://github.com/CFougeres/EVASIONScode>. The code is briefly described in Supplementary information."

8. line 386: more detail is needed than "several criteria"

The sentence has been changed as follows.

"SHIVA and MESA use two basic criteria for the ejection of a specific layer: if its velocity achieves the escape velocity ($\sim 1000 \text{ km s}^{-1}$), or if its luminosity becomes higher than the Eddington limit (when the force exerted by radiation exceeds the gravitational pull) and hence, the condition of hydrostatic equilibrium no longer holds."

Reviewer #3:

The data given and the results and conclusions regarding the ^{22}Na gamma-ray flux derived from that data appear to be both valid and relatively robust. While the variation in WD mass, accretion rate, and WD luminosity give the reader an understanding of the uncertainty in the expected ^{22}Na flux due to the stellar parameters, that uncertainty due to the reaction rate uncertainty is not given, which would be useful in ascertaining the relative effects of these two types of uncertainties on the flux. Presumably both of these will affect the uncertainty in the final determination of the maximum detectability distances for future gamma-ray missions, which should be included (or at least the contribution of the reaction rate uncertainty given the focus of this manuscript on that new determination).

Authors:

We would first like to briefly clarify an issue related to the concept of "uncertainty". In our opinion, the word uncertainty defines well what happens with some reaction rates: while a single value of the rate is expected for a given set of thermodynamic conditions (i.e., specific values of T and density), the lack of precise knowledge of certain nuclear physics inputs prevents us from deriving a precise value of the rate. However, the situation described by the referee, regarding different values for the white dwarf mass or the mass-accretion rate can not, in our opinion, be treated as an uncertainty. It simply reflects the fact that astrophysical

objects, with different properties, can produce different amounts of ^{22}Na . Even if all astrophysical objects could be described with 100% certainty in terms of mass, radius, luminosity, etc. there will be a variation in the possible ^{22}Na yields, since they can be produced in different nova systems.

Following the referee's suggestion, the uncertainty in the maximum detectability distance has been quantitatively estimated. We have computed 2 identical nova models of 1.15 Msun (a representative case, model 115b Table 2), one using the new upper limit to the $^{22}\text{Na}(p,\gamma)^{23}\text{Mg}$ rate and the second with the lower limit to the rate. The mean ^{22}Na yield in the ejecta obtained in the two models is $X(^{22}\text{Na}) = 2.73 \times 10^{-4}$ and 3.81×10^{-4} , respectively, which corresponds to an uncertainty of a factor $F = \text{higher/lower} = 1.4$ in the yield, for this particular model. This translates into an uncertainty of a factor $\sqrt{F} = 1.18$ in the maximum detectability distance. It is worth noting that while the exact uncertainty will depend on the specific inputs used in the model (white dwarf mass and initial luminosity, composition and rate of mass transfer from the companion star), the value obtained (18 %) can be regarded as representative of the uncertainty expected in the maximum detectability distance.

The relevant change has been made on page 11.

“With the expected sensitivities of e-ASTROGRAM (3×10^{-6} photons $\text{cm}^{-2} \text{s}^{-1}$) and COSI (1.7×10^{-6} photons $\text{cm}^{-2} \text{s}^{-1}$) at 1 MeV, assuming model 125 (Table 2), new maximum detectability distances of 2.7 kpc and 4.0 kpc, respectively, have been derived, on which the reaction rate uncertainty obtained here leads to an uncertainty of 18%”

While gamma-spectroscopy is outside my area of expertise, in general the methodology seems sound, and the experimental equipment used is well understood. Perhaps the most uncertain part of the analysis lies in the simulations of the target profile and the stopping power of the ^{23}Mg in the Au foil. Tabulated stopping powers are well-known to be somewhat imprecise; however, the authors have used relatively conservative uncertainties of 20% for these tables in their error budget, which is the norm. In the supplementary information, it is stated that these simulations/analysis tools have been validated by comparisons to known states. Providing more information here (which states were used if validation was done as part of this study, or a reference if one exists) would give the reader more confidence in these results.

With respect to your comment, the following has been added in the Supplementary information: “Certain parameters, such as energy losses in the target, are not known precisely. A systematic uncertainty on these parameters has thus been added. For the states at $E_x = 5292.0(6)$ keV and $E_x = 3796.0(1.2)$ keV, the lifetimes measured in this work, $4(+1-3)$ fs and $40(+6-7)$ fs respectively, are in agreement with those reported in literature of $5(2)$ fs and $41(6)$ fs, thus validating the simulation and analysis tools.”

In addition to the comments provided above, the following edits would improve the clarity and accuracy of the manuscript:

- The value quoted for the resonance strength determined by Ref. [13] in the manuscript is given as 1.4(3) meV; however, Stegmüller et al., give that value for the contribution to the resonance strength for the specific transition to the 451-keV state. Admittedly, this is given as a 100% branch by Jenkins et al., but nevertheless their recommended value for the resonance strength and the value used in their reaction rate calculations is given in their Table 1 as 1.8(7) meV. For consistency of comparing reaction rate results, the authors may wish to use this latter value.

It has been changed to 1.8(7) meV.

- Reference [6] is given for the statement that the ejected mass of ^{22}Na in novae has been shown to be inversely proportional to the lifetime of the 7785-keV state; however, there is no discussion of the lifetime of this state in that publication.

The measured strength is “higher by a factor of 3.2”, and “the amount of ^{22}Na produced is reduced by a factor of 2-3 compared to the previous rate, depending on the nova model” is given in Ref. [6]. It is based on this, that the ejected mass of ^{22}Na is inversely proportional to the lifetime was mentioned. This sentence has been improved.

“Our simulations show that, in the range of the debated values of the resonance strength, the mass of ^{22}Na ejected in novae depends on the lifetime (τ) of this key state approximately as $M_{\text{eject}} \sim \tau^{0.7}$.”

- In lines 160 and 200 “rest energy” of the gamma ray is presumably referring to the center of mass energy? If so, using the latter terminology might help alleviate confusion for a more junior reader who may incorrectly interpret this as meaning rest mass.

This has been changed suitably “...of the center-of-mass energy $E_{\gamma,0}$...”

- Is the shaded red region shown in Fig. 2 (right) the 1-sigma uncertainty of the simulation or the data? The caption should be amended to make this clear to the reader.

The text below has been added:

“The red shaded area corresponds to the simulations with lifetimes within 1 σ uncertainty.”

- Is there a source that can be cited for the statement that for short lifetimes the deceleration force is constant leading to a Delta-beta that is proportional to delta-t? Why this should be so is not immediately clear from the text.

We have $ma=dE/dx \sim \text{constant}$, with m the mass of the ions and a the deceleration. We added in the text: “(dE/dx \sim constant)” and the word “force” was removed.

- It seems that a temperature value may be missing in line 286.

We added the text: “See Table 2”

• While the angular distributions should not affect the lifetime determination as stated in the paper, those likely should be included in the determination of the branching ratio. It is not clear from that section of the Methods part of the manuscript if this was taken into account.

This is corrected in section 3 of the *Supplementary information*, section about the Numerical Simulations. We added the text: “All particle emissions are isotropic. Note that protons emitted from the key state have $\neq 0$.”

• In the supplemental material the authors compare their new reaction rate to those of Sallaska et al. and Stegmüller et al., and give the uncertainty of their newly determined rate. Including both the uncertainties in these previous determinations, as well as a comparison between this result and the recent reaction rates from Jenkins et al. and Friedman et al. would provide a more complete picture of the improvements due to this study compared to the literature.

This has been corrected in the *Supplementary information* by adding the reaction rate from *Jenkins et al.* in Fig. 4 (green curve). We note that *Friedman et al.* did not publish a reevaluated rate. For completeness, we added the following sentence: “The uncertainties in the recommended rate result in an uncertainty of a factor of 1.4 in the estimated yield of ^{22}Na when a representative case (i.e., Model 125, Table 2 in the main text) is adopted. For comparison, the uncertainties of previous yield estimations were of a factor of 4 and 3.25”

Additional corrections:

- Acknowledgement. This work was supported by the ASTRANUCAP IRN, NuAG IRP, SPIRAL2-CZ, and the U.S. Department of Energy, Office of Science, Office of Nuclear Physics, under contract number DE-AC02-06CH11357. This work was also partially supported by the MINECO grants PID2020-117252GB-I00 and PID2020-118265GB-C41, and by the AGAUR/Generalitat de Catalunya grant SGR-386/2021. V.G. would like to thank São Paulo Research Foundation (FAPESP) (grant 2016/02863-4) and CNPq (grant 302969/2013-6).
- Addition of one author (A. Gottardo).
- Ref. 24 (Fougeres et al., EPJ Web Conf.) from accepted to published.
- Nature Communications formatting (abstract length, header etc.) following the recommendations of the Editor.
- The source data of the figures are provided with this paper (indicated in “Data Availability” section).
- Several minor language improvements have been made to the text and are indicated in red.

REVIEWER COMMENTS

Reviewer #1 (Remarks to the Author):

There are no comments on the revised manuscript.

Reviewer #2 (Remarks to the Author):

None, I think the authors have addressed my (and the other reviewers' questions) appropriately. I recommend publication.

Reviewer #3 (Remarks to the Author):

The authors of “Search for ^{22}Na in novae supported by a novel method for measuring femtosecond nuclear lifetimes” have adequately addressed most of the comments/concerns raised in the original review. While the new values for lifetime and proton branching ratio are effectively the same as previously accepted values, the results are nevertheless relevant for multiple fields. These findings will have a strong impact on both nuclear physics and astrophysics, with the new method for measuring femtosecond lifetimes of states described herein and the predictions of the potential observations of ^{22}Na from novae by future missions, respectively. There are a few additional comments that the authors may wish to address before publication that were noticed upon this second review and are enumerated below.

Comparing Figs. 3a and 3b in the Supplementary Information, the counts in the region where the proton decays of the state of interest should be (i.e. 7.5-8 MeV on both axes) do not seem to be above the background rate to the left of the red line. The text describes a background subtraction from a run on a pure gold target; however, is there any background subtraction of the data completed the constant background shown in the actual spectrum? Relatedly, the center-of-mass energy of the proton decays from the state of interest is only ~ 300 keV and thus their lab energy deposited in the detector would presumably be close to the detector threshold. Is the energy threshold of the detector well understood?

One assumes that the two gold foils (the implanted one and the stopping one) were back-to-back? It may be helpful to the reader to say this explicitly in the text.

Given the discrepancies between excitation energies given in the most recent evaluation (Ref. [16]) and those given in the text, one assumes that the values given here are those measured in this experiment, specifically, as determined by gamma-ray transitions detected by AGATA as their uncertainties are quite small. The authors should state this explicitly near the beginning of the Results section.

Ref. [22] constrains the spin of the state of interest to be $3/2+$ or $5/2+$ but only the latter is given in the text. These ($3/2+$ and $5/2+$) are also the potential spins listed by the most recent evaluation [16] with no mention of a potential $7/2+$ spin.

One might take exception with the statement in lines 207-208 that the excitation energy is precisely selected by VAMOS given that the FWHM resolution is 500 keV.

In lines 260-261 it would be useful to include that previous uncertainties of the reaction rate such that the reader knows how much they have been reduced.

Point-by-point response to the reviewers' comments to the Authors, reproduced verbatim

Please find the responses to the referees' comments and queries.

Reviewer #3 (Remarks to the Authors):

Comparing Figs. 3a and 3b in the Supplementary Information, the counts in the region where the proton decays of the state of interest should be (i.e. 7.5-8 MeV on both axes) do not seem to be above the background rate to the left of the red line. The text describes a background subtraction from a run on a pure gold target; however, is there any background subtraction of the data completed the constant background shown in the actual spectrum?

Figure 3 (a) is a matrix where the background has been subtracted. The error bars in Figure 3 (c) and (d) take into account the error associated with this background subtraction.

We have made this clear in the caption with this sentence:

“The background measured with a pure gold target has been subtracted.”

Relatedly, the center-of-mass energy of the proton decays from the state of interest is only ~300 keV and thus their lab energy deposited in the detector would presumably be close to the detector threshold. Is the energy threshold of the detector well understood?

Most of the emitted protons have energies well above the detector threshold. This is because of the kinematic boost of the protons (inverse kinematics) which transforms the ~300 keV center-of-mass energy into up-to-3-MeV energy in the laboratory.

One assumes that the two gold foils (the implanted one and the stopping one) were back-to-back? It may be helpful to the reader to say this explicitly in the text.

This following text has now been added:

“The two gold foils were mounted back-to-back. “

Given the discrepancies between excitation energies given in the most recent evaluation (Ref. [16]) and those given in the text, one assumes that the values given here are those measured in this experiment, specifically, as determined by gamma-ray transitions detected by AGATA as their uncertainties are quite small. The authors should state this explicitly near the beginning of the Results section.

We added in the text:

“(we use energy values from the present measurement throughout this article)”

Ref. [22] constrains the spin of the state of interest to be $3/2+$ or $5/2+$ but only the latter is given in the text. These ($3/2+$ and $5/2+$) are also the potential spins listed by the most recent evaluation [16] with no mention of a potential $7/2+$ spin.

The $7/2+$ spin is discussed in all articles. Even in reference [22], $5/2+$ and $7/2+$ were adopted in the rate estimation, and it is a mistake that $7/2+$ is not in reference [16]. The arguments given for $7/2+$ are still valid. We changed the text:

“... three values have been proposed: $J=3/2+$, $5/2+$ or $7/2+$...”

One might take exception with the statement in lines 207-208 that the excitation energy is precisely selected by VAMOS given that the FWHM resolution is 500 keV.

The word “precisely” was removed.

In lines 260-261 it would be useful to include that previous uncertainties of the reaction rate such that the reader knows how much they have been reduced.

This is presented in the Supplementary Information (Fig. 4). As discussed in the main text, this new rate was found to be very reliable at maximum nova temperatures, with uncertainties reduced to 40% (10%) at $T=0.1\text{GK}$ (0.5GK). For example, this can be compared with Jenkins 2004, see Fig 4, where it is 200% (50%).

We added in the main text:

“(see Supplementary information)”

REVIEWERS' COMMENTS

Reviewer #3 (Remarks to the Author):

The authors have adequately responded to all comments/suggestions given in the two reviews provided. Their results are of broad interest to both the nuclear and astrophysics communities and represent progress in the attempt to understand the observation (or lack thereof) of gamma-ray observations from ^{22}Na by future missions.

Point-by-point response to the reviewers' comments to the Authors, reproduced verbatim

We would like to thank the reviewers for their questions and comments and their help in improving the overall quality of the article.